# Regulation of Polyamine Metabolism by Curcumin for Cancer Prevention and Therapy

**DOI:** 10.3390/medsci5040038

**Published:** 2017-12-18

**Authors:** Tracy Murray-Stewart, Robert A. Casero

**Affiliations:** Johns Hopkins University, Sidney Kimmel Comprehensive Cancer Center, Baltimore, MD 21287, USA; tmurray2@jhmi.edu

**Keywords:** curcumin, diferuloylmethane, ornithine decarboxylase, polyamine, NF-κB, chemoprevention, carcinogenesis, polyphenol

## Abstract

Curcumin (diferuloylmethane), the natural polyphenol responsible for the characteristic yellow pigment of the spice turmeric (*Curcuma longa*), is traditionally known for its antioxidant, anti-inflammatory, and anticarcinogenic properties. Capable of affecting the initiation, promotion, and progression of carcinogenesis through multiple mechanisms, curcumin has potential utility for both chemoprevention and chemotherapy. In human cancer cell lines, curcumin has been shown to decrease ornithine decarboxylase (ODC) activity, a rate-limiting enzyme in polyamine biosynthesis that is frequently upregulated in cancer and other rapidly proliferating tissues. Numerous studies have demonstrated that pretreatment with curcumin can abrogate carcinogen-induced ODC activity and tumor development in rodent tumorigenesis models targeting various organs. This review summarizes the results of curcumin exposure with regard to the modulation of polyamine metabolism and discusses the potential utility of this natural compound in conjunction with the exploitation of dysregulated polyamine metabolism in chemopreventive and chemotherapeutic settings.

## 1. Introduction

Chemoprevention entails the long-term use of synthetic or natural agents by healthy individuals, particularly those with a predisposing cancer risk, to delay disease onset. As such, potential side effects and off-target effects must be absolutely minimal. Natural products derived from foods are therefore at an advantage due to their accessibility and history of safe consumption. Epithelial cancers are often age-related cancers: through its long-term, direct interaction with environmental and dietary factors, the epithelium has the greatest potential for interactions that might prevent or modulate the course of tumorigenesis. Naturally, gastrointestinal (GI) cancers have one of the greatest potentials for dietary factor influence. As approximately 20% of cancers worldwide are associated with infection or inflammation [1,2], the anti-inflammatory and antioxidant properties associated with many natural products might be of particular value. Nutritional components also have the potential to participate in therapeutic strategies, and elucidating the molecular mechanisms of these agents, including traditional medicines, is providing clues as to how they might best be incorporated into treatment regimens.

## 2. Polyamines and Cancer 

Increases in polyamine biosynthesis and intracellular polyamine content are some of the most consistent biochemical alterations observed in cancer cells of all types, indicating their importance in tumorigenesis [3,4]. The mammalian polyamines include spermine, spermidine, and putrescine, which are essential polycations with pleiotropic roles in cellular proliferation and survival (Figure 1) [3]. Due to their positive charge at physiological pH, many of the essential functions of polyamines stem from their interactions with negatively charged cellular components, including DNA, RNA, certain proteins, and ion channels [5,6,7,8].

In neoplastic cells, loss of polyamine homeostasis occurs and is accompanied by dysregulated proliferation involving upregulated biosynthesis, downregulated catabolism, and increased uptake (Figure 2) [4,9,10,11]. The activity of ornithine decarboxylase (ODC), the initial rate-limiting step in polyamine biosynthesis, has been directly correlated with the rates of DNA synthesis and cellular proliferation in multiple tissue types [3].

The requirement for polyamines increases over the course of tumorigenesis, and studies in multiple human cancer types have demonstrated an elevation of ODC activity and/or polyamines in neoplastic or tumor tissue relative to adjacent normal tissue [12,13,14,15]. In a cohort of 50 primary breast tumors, the level of ODC activity demonstrated a strong negative correlation with both disease-free and overall survival, indicating ODC as a poor prognostic factor [16]. Polyamines have been implicated in oncogenic and viral transformation, and ODC activity is rapidly induced upon exposure to oncogenic or growth-promoting stimuli [10,17,18,19]. In particular, polyamine biosynthesis is upregulated at multiple steps by the *c-MYC* oncogene [20,21,22], and the activation of *k*-*RAS* rapidly induces ODC activity to promote malignant transformation and oncogenesis [23,24,25]. Colorectal carcinoma biopsies with activating *k-RAS* mutations were shown to have enhanced polyamine biosynthesis compared to those with wild-type *k-RAS* [26]. The expression level of ODC has been shown to directly correlate with the potential to promote tumorigenesis in both lymphomas and in solid tumors [27,28]. Furthermore, ODC activity rapidly increases with exposure to chemical carcinogens or tumor promoters, and this elevated expression is often utilized as a biomarker of tumor promotion in carcinogenesis models [3,29,30,31].

### 2.1. Targeting Polyamine Metabolism for Cancer Prevention

The potential targeting of polyamine biosynthesis as an antiproliferative strategy came with the recognition that elevated polyamine biosynthesis was a general requirement for the survival of cancer cells [32,33,34]. In fact, the ability of an agent to inhibit ODC activity is commonly considered a predictor of chemopreventive activity [14]. The most widely studied and successful inhibitor of polyamine biosynthesis, α-difluoromethylornithine (DFMO), or eflornithine, is enzyme-activated and irreversibly inhibits ODC through covalently binding with its active site [35]. DFMO typically elicits cytostatic effects in cell culture models through the depletion of putrescine and spermidine, with variable effects on spermine, and it is capable of preventing tumor formation in numerous animal models [36]. Early clinical trials investigating the prevention of colorectal cancer using low doses of DFMO have established the safety of its administration as well as its efficacy in reducing polyamine levels in colorectal mucosa [37].

Subsequent studies in colorectal cancer models have described enhanced antitumor benefits when combining DFMO with common non-steroidal anti-inflammatory drugs (NSAIDs), including sulindac and celecoxib. In addition to DFMO inhibiting ODC activity, the addition of an NSAID further decreased intracellular polyamine content by stimulating polyamine catabolism and export through activation of spermidine/spermine-*N*^1^-acetyltransferase (SSAT), resulting in an additive reduction in the formation of colon tumors [38,39,40]. The combination of DFMO and sulindac has since been clinically investigated with impressive outcomes including a 70% reduction in the number of total metachronous colorectal adenomas [41,42], and additional studies are ongoing. Clinical trials have also been conducted investigating the efficacy of DFMO as a chemopreventive agent in individuals with a history of non-melanoma skin cancer, actinic keratosis (a squamous cell carcinoma precursor), Barrett’s esophagus, and prostate cancer (reviewed in [43]). Although mostly in early clinical phases, DFMO was safely administered in each of these studies, and the results warranted further trials. Importantly, these studies have provided strong evidence supporting the targeting of polyamine metabolism as a valid strategy for the prevention of tumorigenesis, particularly in those susceptible to colorectal cancer.

It should be noted that the long-term use of DFMO is not without minor, but unwanted, side effects: a dose-related, but generally reversible, ototoxicity has frequently occurred in patients on long-term treatment. However, a reduced daily oral dose of DFMO has been shown to be sufficient in reducing polyamine levels in the colorectal mucosa while minimizing these side effects [31]. Furthermore, combination therapies, such as that with DFMO and NSAIDs, allow for lower doses of the individual agents, thereby lessening the risk for side effects. This is also important for the long-term use of NSAIDs as chemoprevention, as the most common side effects are gastrointestinal mucosal injury and renal toxicity, with cardiovascular, central nervous system (CNS) and platelet side effects occurring less frequently [44].

### 2.2. Targeting Polyamine Metabolism for Cancer Treatment

Although DFMO effectively inhibits cellular proliferation in a variety of cancer cell types in vitro and in vivo, its efficacy as a monotherapy against established tumors in clinical trials has been mostly unsuccessful due to compensatory uptake of polyamines released into the intestinal lumen through the turnover of gut mucosal cells as well as the microbiome and diet (reviewed in [45,46]). Polyamine analogues have thus been developed with the ability to downregulate polyamine biosynthesis through negative feedback mechanisms, inhibit uptake through the polyamine transporter, and induce the catabolism of the natural polyamines [47,48,49]. These structural mimetics are capable of competing with natural polyamines for binding sites, but are unable to substitute for their growth-sustaining functions. Treatment of a wide variety of cancer cell lines both in vitro and in xenograft mouse models with members of the symmetrically substituted bis(ethyl)polyamine analogues has resulted in tumor-specific cytotoxicity that is associated with depletion of the natural polyamines and, in some cases, generation of reactive oxygen species (ROS) [33,34,47,48]. Early clinical trials with a second-generation analogue, PG-11047 [50], have shown it to be well tolerated as both a single agent and in combination with common chemotherapeutic agents (clinicaltrials.gov #NCT00293488, NCT00705653, NCT00705874). Most recently, two of these analogues, bis(ethyl)norspermine (BENSpm) and PG-11047, were used to generate polycationic polymers capable of targeting polyamine catabolism while simultaneously acting as nanocarriers for the delivery of therapeutic nucleic acids, including microRNA (miRNA) and small interfering RNA (siRNA) [51,52].

## 3. Dietary Factors with Potential to Moderate Carcinogenesis through the Polyamine Pathway

Although drug development targeting the polyamine pathway is progressing, evidence also suggests that many naturally occurring compounds already present in our diet can affect polyamine metabolism, with ultimate effects on cancer prevention or treatment. The interaction with dietary components is greatest in the gastrointestinal tract, and it is here that the anticancer potential of dietary or environmental factors might be most advantageous. As with DFMO and NSAIDs, colorectal cancer (CRC) model systems have historically been used to study the efficacy of both synthetic and naturally occurring dietary agents, including plant polyphenols, phytoestrogens, and probiotics, with regard to chemopreventive and chemotherapeutic potential.

### 3.1. Plant Polyphenols

Naturally occurring polyphenols include several subclasses of structurally related plant substances long recognized for their health benefits in traditional medicine. They can be classified as phenolic acids (highly concentrated in coffee, teas, pomegranate, and berries), stilbenes (including resveratrol; found in grapes, wine, and blueberries), tannins (grapes, tea, coffee, lentil, and walnuts), diferuloylmethanes (turmeric), and flavonoids, which constitute the largest subclass of phenolic compounds and are the major source in the average diet. Many flavonoids are produced by plants as a means of protection against parasites, oxidative injury and harsh environmental stress conditions, and can be further classified into groups including, but not limited to, anthocyanins (blue- or purple-pigmented fruits), flavanols (including catechins found in teas, dark chocolate, and cocoa), flavanones (citrus fruits) and isoflavones (phytoestrogens in soy products, such as genistein). Representative compounds of nearly all of these groups have been demonstrated to affect the polyamine metabolic pathway, primarily through an inhibition of ODC activity, resulting in decreased tumorigenesis. The impact of certain flavonoids, including resveratrol, genistein, and green tea (-)-epigallocatechin-3-gallate (EGCG) on polyamine metabolism and colorectal carcinogenesis, as well as in other carcinogenesis models, is evident, and these data have been comprehensively reviewed by Russo and colleagues [53,54]. The remainder of the current review will therefore focus only on curcumin and its potential in the regulation of polyamine metabolism.

### 3.2. Curcumin

Curcumin, or 1,7-bis(4-hydroxy 3-methoxy phenyl)-1,6-heptadiene-3,5-dione, is a naturally occurring polyphenol that has been the focus of many studies in a variety of medical fields (Figure 3). A phenolic compound, this yellowish orange pigment’s only source is the rhizomatous turmeric plant (*Curcuma longa*), which is cultivated mostly in India and Southeast Asia. Curcuminoids constitute only approximately 5% of turmeric root powder, and exist in 3 forms: curcumin, also referred to as diferuloylmethane (60–70% of crude extract), desmethoxycurcumin (20–27%), and bisdesmethoxycurcumin (10–15%) [55].

Commonly used as a spice and food coloring as well as in skin care products and textile dyes, curcumin has been used for centuries in traditional Chinese medicine and Indian Ayurvedic medicine for its health-promoting properties. The pathologies believed to benefit from curcumin are diverse, and likely stem from its potential to regulate key molecular processes involved in the pathology of many diseases. Of particular interest in cancer etiology and prevention are the antioxidant, anti-inflammatory, and antiproliferative benefits attributed to dietary curcumin. As an antioxidant, curcumin can act as a free radical scavenger, inhibit the generation of free radicals and subsequent oxidative damage, and induce the activity of antioxidant molecules and enzymes involved in detoxification processes, such as glutathione-*S*-transferase (GST) and nuclear factor E2-related factor (NRF-2) [56,57].

The role of curcumin in the anti-inflammatory response is associated with its ability to downregulate certain transcription factors that promote the production of inflammatory gene products. Perhaps most significantly, curcumin inhibits nuclear factor-kappa B (NF-κB) activation, preventing its translocation into the nucleus where it could directly induce the transcription of genes associated with cell survival and inflammation [58,59]. Affected genes include the free-radical-producing enzymes cyclooxygenase-2 (COX2), lipoxygenase (LOX), and inducible nitric oxide synthase (iNOS) as well as pro-inflammatory cytokines, such as interferon gamma (IFNγ), tumor necrosis factor alpha (TNFα), interleukin (IL)-1, IL-6, and others. In addition, curcumin invokes growth inhibitory effects through other NF-κB target gene products, including cyclin D1 and c-MYC, induces apoptosis in cancer cells, and demonstrates anti-angiogenic activity. Overall, the importance of curcumin in cancer chemoprevention and treatment may originate from its inhibitory effect on molecules linking inflammation and cancer. As several genes encoding polyamine metabolic enzymes are also regulated by many of the above-mentioned molecules, the potential exists for transcriptional modulation of the polyamine pathway via curcumin.

## 4. Investigations into the Antitumor Potential of Curcumin through Modulating Polyamine Metabolism

### 4.1. Evidence of the Chemopreventive Activity of Curcumin in Carcinogenesis Models

Animal models of carcinogenesis involve the administration of carcinogens and/or toxicants that act as tumor initiators or promoters. Studies with DFMO established a critical role for ODC induction by tumor promoters such as phorbol esters in the early stages of tumor development [60]. Subsequently, ODC has been used as an indicator of tumor promotion induced by a variety of agents in multiple carcinogenesis model systems [31].

#### 4.1.1. Topical Application of Curcumin in Animal Models of Skin Cancer

The effects of curcumin on polyamine metabolism were first investigated in the CD-1 skin carcinogenesis mouse model [61]. The tumor promoter 12-*O*-tetradecanoylphorbol-13-acetate (TPA) is well established to rapidly induce ODC activity [30], and this ODC induction contributes to tumorigenesis [3]. Topical application of curcumin to the epidermis concurrently with TPA potently inhibited the induction of epidermal ODC activity in a dose-dependent manner. Similarly, TPA-induced DNA synthesis was progressively inhibited by increasing doses of curcumin. Ultimately, in a two-stage initiation-promotion model using 7,12-dimethylbenz[*a*]anthracene (DMBA) followed by TPA, curcumin potently reduced TPA-induced tumor promotion, resulting in a 98% decrease in the number of skin tumors observed [61]. In a similar strategy, both topical and intraperitoneal (i.p.) administrations of curcumin were investigated for their effects on TPA-induced ODC mRNA and activity in mouse epidermis. Both routes of administration were capable of inhibiting the induction of ODC mRNA and activity in a near-parallel, dose-dependent manner, indicating that modulation of ODC by curcumin occurs primarily at the mRNA transcript level [62].

The results of these studies were further verified in vitro using ME308 mouse keratinocytes established from DMBA-initiated mouse skin [63]. Lee and Pezzuto [64] used this system to investigate an extensive panel of potential chemopreventive agents, including curcumin, for their ability to inhibit TPA-induced ODC activity. Their data revealed that co-incubation of curcumin with TPA provided one of the most potent inhibitory effects on ODC activity, with a half maximal inhibitory concentration (IC_50_) of 4 μM. In comparison, the irreversible ODC inhibitor DFMO prevented TPA-induced ODC activity with an IC_50_ of 20 μM. A similar inhibitory effect for curcumin compared to DFMO was obtained in a subsequent screen using the rat 2C5 tracheal epithelial cell line to investigate TPA-induced ODC activity [65]. Of note, curcumin was not toxic to this non-tumorigenic immortalized cell line.

Studies into the mechanism of tumor promotion by TPA revealed a critical role for protein kinase C (PKC) signaling in mediating many TPA-induced tumorigenic effects, including ODC activity. Topical application of curcumin to the dorsal skin of Swiss bare mice prevented TPA-induced PKC translocation, resulting in effects analogous to those observed using a known selective inhibitor of PKC. These effects of PKC inhibition ultimately included the repression of ODC induction, ROS generation, apoptosis, and hyperplasia, which were associated with alterations in TPA-induced kinases and transcription factors [66].

Irradiation of CD-1 mice with ultraviolet A (UVA) has been shown to enhance the tumor-promoting effects of TPA on the epidermis beyond that observed with TPA alone. These effects include increased ODC activity and dermatitis, as evidenced by dermal infiltrating inflammatory cells, and topical pretreatment with curcumin significantly prevented these increases as well as those observed with TPA alone [67]. In the same model, it was subsequently determined that although ODC mRNA was induced by TPA, as previously reported [62], it was not significantly enhanced by addition of UVA; ODC activity was, however, induced by UVA in addition to TPA, suggesting post-transcriptional regulation of ODC by UVA. Importantly, pretreatment with curcumin could block the UVA-TPA-stimulated induction of both ODC mRNA and protein [68].

#### 4.1.2. Dietary Curcumin in Rodent Models of Carcinogenesis

Following the initial finding that curcumin prevented TPA-induced ODC activity and tumorigenesis when applied topically, studies were conducted investigating the effects of dietary administration of curcumin. At the time of these studies, the chemopreventive effect of NSAIDs on colon tumorigenesis was becoming apparent. As curcumin has traditionally been used in the treatment of a variety of inflammatory conditions, its potential for inhibiting colon tumorigenesis was investigated. F344 rats were fed diets containing 2000 p.p.m. curcumin for two weeks prior to subcutaneous injections of azoxymethane (AOM), a carcinogen that specifically induces distal colon tumors in rodents with a pathology mimicking that of sporadic human colon cancers [69]. In the group receiving the curcumin-supplemented diet, AOM-induced ODC activity was significantly decreased in the colonic mucosa as well as the liver, where AOM is metabolically activated. Additionally, animals receiving dietary curcumin prior to AOM exposure demonstrated a greater than 50% reduction in the number of AOM-induced aberrant crypt foci (ACF), which are early preneoplastic lesions in the colon [70].

The F344 rat strain was also used to investigate the protective effect of curcumin on 4-nitroquinoline 1-oxide (NQO)-induced oral carcinogenesis. 4-NQO is easily administered to rats in the drinking water and produces tongue lesions including squamous cell neoplasms that are comparable to those in human oral carcinogenesis [71]. To study the efficacy of curcumin in preventing the initiation of tumorigenesis, rats were fed a diet containing 500 p.p.m. curcumin starting one week prior to and throughout 4-NQO exposure for 8 weeks. A second group received no curcumin until one week after the 8-week 4-NQO exposure (post-initiation), and curcumin remained in the diet of this group until the conclusion of the study 22 weeks later. Rats in both curcumin groups demonstrated impressive reductions in the frequency of tongue neoplasms: 4-NQO-induced carcinomas were reduced from 54% of the animals to 5% when curcumin was added at the initiation phase and to 15% when added post-initiation. The number of animals with preneoplastic squamous cell dysplasias was similarly decreased. Analyses of the polyamine content of normal-appearing sections of tongue tissue at the end of the study revealed that 4-NQO exposure significantly elevated the levels of spermidine and spermine, as well as total polyamine content, above that of untreated animals, consistent with an induction of ODC activity. Importantly, these elevated levels were prevented when curcumin was also administered, with initiation phase exposure providing the greatest benefit and maintaining polyamine pools that did not significantly differ from those of rats not receiving carcinogen. These reduced polyamine levels and carcinoma numbers were also accompanied by reductions in proliferation biomarkers in the tongue epithelium [72]. Of note is a previous study conducted by the same investigators that analyzed the effects of the ODC inhibitor DFMO on NQO-induced oral carcinogenesis with very similar results [73].

In a third model system, male ddY mice were pretreated with 1% dietary curcumin for 4 weeks prior to receiving i.p. injections of the renal carcinogen ferric nitrilotriacetate (Fe-NTA). Twelve hours after administration of Fe-NTA, ODC activity levels in the mouse kidney were increased approximately 4.4-fold, while pretreatment with curcumin inhibited this increase by 63%, with no effect on basal enzyme activity. Concurrently, Fe-NTA-generated oxidative stress, a mechanism associated with its tumor-promoting abilities, and nephrotoxicity were also alleviated by curcumin pretreatment [74].

### 4.2. Investigations into Polyamine-Associated Effects of Curcumin on Established Tumors—Potential for Cancer Treatment

Unlike non-tumorigenic cells, cancer cells typically respond to curcumin exposure through inducing apoptosis and cell death [75]. Several characteristics of cancer cells are responsible for this differential sensitivity, including increased curcumin uptake and ROS generation, lower glutathione levels, and the constitutive activation of NF-κB that often mediates the survival of cancer cells [75,76,77,78]. In addition to being a c-MYC-regulated gene, evidence exists for the regulation of ODC by NF-κB [79], and the SSAT catabolic enzyme is also an NF-κB target [80]. Studies have also suggested that NF-κB can be activated by the elevated levels of polyamines present in tumor cells [81]. Thus, the inhibition of polyamine biosynthesis by curcumin might indirectly inhibit NF-κB activation. In spite of these potential mechanistic links in terms of regulation by transcription factors, relatively few studies have investigated the effect of curcumin exposure on polyamine metabolism in cell lines. Mehta et al. [82] first analyzed the effect of curcumin on a panel of 8 breast cancer cell lines representing multidrug-resistant (MDR), estrogen-dependent, and estrogen-independent breast cancers. Impressive growth inhibitory effects were observed in all of the cell lines (1–26% viability following 1 μg/mL treatment for 72 h), including an Adriamycin-resistant MCF-7 line (~15% viability). This antiproliferative activity was time- and dose-dependently correlated with curcumin-induced inhibition of ODC activity. Interestingly, in MCF-7 cells, there was no apparent evidence of apoptosis and the apoptosis-related genes examined remained unchanged [82]. Flow cytometric studies in MDA-231 cells indicated temporary growth arrest at the G2/M checkpoint after 24 h of curcumin exposure; however, cells appeared to re-enter the cell cycle with longer exposure times. The results of this study were the first to suggest the potential use of curcumin as an antiproliferative agent against breast cancer cells.

In a more recent study using SK-BR-3 breast cancer cells, which overexpress Human Epidermal Growth Factor (HER)-2, curcumin inhibited growth, and a flow cytometric assay measuring bromolated deoxyuridine triphosphate (BrdU) incorporation indicated the induction of apoptosis even at the lowest concentration examined (2.5 μM after 72 h). Analysis of intracellular polyamine pools following curcumin treatment revealed substantial decreases in spermidine and spermine levels (85% and 50%, respectively), suggesting that the loss of intracellular polyamines might be important in the antiproliferative mechanism of curcumin [83].

Curcumin inhibits the activation and nuclear translocation of NF-κB, thereby preventing stimulation of NF-κB target genes [58]. The involvement of NF-κB signaling in curcumin-mediated changes in polyamine metabolism was recently investigated in human breast cancer [84]. Pretreatment of MCF-7 breast cancer cells with an inhibitor of NF-κB prior to curcumin treatment suggested that curcumin-mediated alterations in c-MYC, ODC, SSAT, and PAOX protein levels occurred in part through this signaling pathway [84]. Interestingly, this response pattern to curcumin was altered by overexpression of the B-cell lymphoma-2 (*BCL2*) gene, an alteration associated with chemo- and radioresistance [85]. Although the MCF-7/BCL2 cells were less sensitive to the effects of curcumin than the parental strain, significant inhibition of colony formation remained evident. Of note, as BCL2 overexpression in itself upregulates NF-κB [86], the basal expression levels of c-MYC, ODC, and SSAT were also substantially increased in this cell line [84], providing further evidence for regulation of these enzymes by NF-κB.

The ability to stimulate apoptosis in cancer cells of various origins is a key anti-carcinogenic property of curcumin. In the HL-60 promyelocytic leukemia cell line, Liao et al. [87] provided comprehensive evidence indicating a role for ODC in the mechanism of curcumin-induced apoptosis. Treatment of these cells with curcumin quickly inhibited ODC enzymatic activity in a time- and dose-dependent manner that correlated with growth inhibition; furthermore, overexpression of ODC or pretreatment of wildtype cells with a caspase inhibitor increased survival in the presence of curcumin. Relative to the parental cell line, ODC-overexpressing HL-60 cells were protected from the apoptotic hallmarks of curcumin treatment and presented little evidence of DNA fragmentation, ROS production, loss of mitochondrial membrane potential or cytochrome *c* release, cleavage of procaspases 3 and 9, downregulation of BCL2, or apoptosis-related morphological changes. Importantly, the loss of curcumin-induced DNA fragmentation observed in the ODC-overexpressing cells was restored by DFMO treatment or siRNA targeting ODC [87].

## 5. Translational Potential, Clinical Trials, and Limitations

In addition to the anecdotal evidence accompanying centuries of traditional medicine, curcumin has been safely administered to humans in many registered clinical trials, with nearly 70 trials completed that targeted a variety of conditions (clinicaltrials.gov). One patient population with potential to benefit from curcumin supplementation includes individuals with familial adenomatous polyposis (FAP), a hereditary form of colorectal cancer resulting from a germ-line mutation of the *adenomatous polyposis coli* (*APC*) gene. ODC activity is elevated in normal-appearing colonic mucosa as well as polyps of patients with FAP [88]; furthermore, pre-symptomatic FAP patients contain elevated colorectal mucosa levels of putrescine [89]. Studies in the Min/+ mouse, a model of FAP with one mutant and one wild type copy of the *Apc* gene, demonstrated a 64% reduction in adenoma formation following daily dietary curcumin intake [90]. In a clinical study of FAP patients, the combination of curcumin and a second polyphenol, quercetin, effectively reduced adenoma polyp number and size; however, treatment arms with the individual agents were not conducted [91]. A recently completed randomized, placebo-controlled phase 2 trial (clinicaltrials.gov identifier #NCT00641147) specifically investigated the effect of daily dietary curcumin supplementation on the regression of adenomas in FAP patients over the course of one year. According to the reported results, no significant benefit was observed for the treatment group in terms of polyp number or size, nor were changes observed in the levels of polyamines, suggesting a lack of drug availability.

The poor solubility, bioavailability, and stability of curcumin are common impediments to its clinical utility, particularly when given orally. However, its stability is increased in acidic environments such as the stomach, and the requirement for systemic bioavailability is lessened with the potential for direct contact. Strategies improving this bioavailability are a current area of research and include such approaches as the use of adjuvants that interfere with the metabolism of curcumin, structural analogues of curcumin, and curcumin-containing nanoparticles [57,92]. The structure of curcumin has been widely modified, with particular focus on changes in the β-diketone structure and aryl substitution pattern of the molecule. Of these structural analogues, the incorporation of a 3,5-dibenzylidenepiperidin-4-one framework elicits enhanced antioxidant and antiproliferative actions relative to curcumin, potentially offering an improved pharmacokinetic profile [93,94,95].

As curcumin has potential therapeutic value against multiple human conditions, enhancing its bioavailability and ascertaining its efficacy in clinical trials could significantly impact the treatment and health of many individuals around the world.

## 6. Conclusions

The ability of curcumin to specifically alter the signaling pathways required for cancer cell survival strongly suggests its potential in chemopreventive and chemotherapeutic strategies, particularly in inflammation- or ROS-associated carcinogenesis. Modulation of polyamine pathway enzymes and the levels of intracellular polyamines appear to contribute to the anticancer potential of curcumin both in terms of carcinogenesis and in the treatment of established tumors. Therefore, establishing the molecular mechanisms underlying the regulation of polyamines by curcumin will potentially add to our understanding of how to most effectively target and prevent tumor cell proliferation and might provide insight on how to best supplement or substitute current more toxic therapies with curcumin.

## Figures and Tables

**Figure 1 medsci-05-00038-f001:**
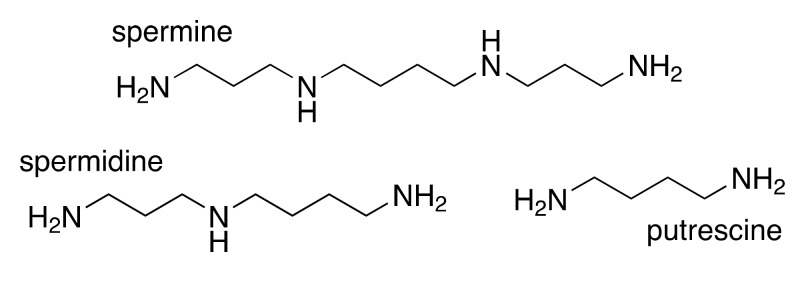
Chemical structures of the primary mammalian polyamines.

**Figure 2 medsci-05-00038-f002:**
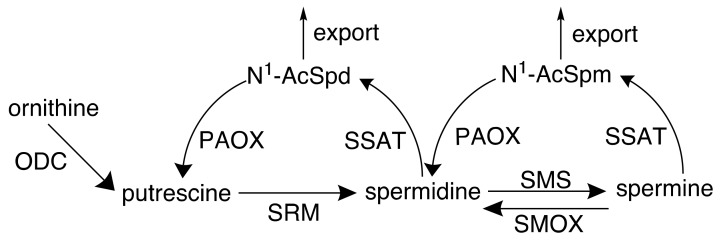
The mammalian polyamine pathway. Polyamines are derived from the amino acid ornithine, which is decarboxylated by ornithine decarboxylase (ODC) to form the diamine putrescine. Putrescine undergoes the sequential addition of 2 aminopropyl groups to form spermidine followed by spermine. These reactions are catalyzed by the spermidine and spermine synthases (SRM and SMS, respectively), using decarboxylated S-adenosylmethionine as the aminopropyl donor. Catabolism of spermine back to spermidine can occur through direct oxidation via spermine oxidase (SMOX) or by acetylation at the N^1^ position by spermidine/spermine *N*^1^-acetyltransferase (SSAT), followed by oxidation by the acetylpolyamine oxidase (PAOX). This latter two-step mechanism also back-converts spermidine to putrescine via an N^1^-acetylspermidine (N^1^-AcSpd) intermediate. Alternatively, acetylated spermine and spermidine can be readily exported from the cell.

**Figure 3 medsci-05-00038-f003:**
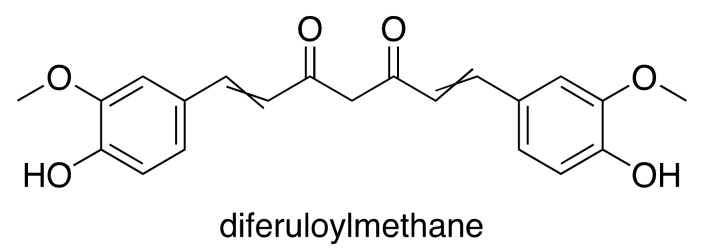
Chemical structure of curcumin, the principle active curcuminoid component of turmeric

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
