# Peer review of "Regulation of Polyamine Metabolism by Curcumin for Cancer Prevention and Therapy"

_medsci, 2017, doi:10.3390/medsci5040038_

Reviewer 1 Report

“Regulation of Polyamine Metabolism by Curcumin for Cancer Prevention and Therapy” is a well-written and thoughtful review of the literature that shows a clear dependency on polyamine metabolism with the cancer chemopreventive activity of curcumin.  A strong case is presented for additional studies to determine how curcumin modulates polyamine metabolism to exert its anti-cancer effects (particularly with regard to inflammation and ROS-associated effects).   The authors suggest that curcumin transcriptionally modulates polyamine metabolism via curcumin-inhibition of NF-κB activation.  However, the authors should also note that it is possible that NF-κB is activated by elevated levels of polyamines that are found in tumor cells [1], and curcumin inhibition of polyamine biosynthesis indirectly inhibits NF-κB activation.

1.  Shah N, Thomas T, Shirahata A, Sigal LH, Thomas TJ., Activation of nuclear factor kappaB by polyamines in breast cancer cells, Biochemistry, 1999, Nov 9;38(45):14763-14774.

Author Response

Thank you for your comments and insightful suggestion to include the possibility of NF-KB regulation indirectly through polyamine levels. This is an important point that further indicates the potential importance of polyamines in the response to curcumin.

The Shah et al. reference has now been included and sentences citing it added in lines 307-309.

Reviewer 2 Report

Reviewer’s Counsels:

Authors need to discuss with emphasis on the detail molecular mechanisms of curcumin chemoprevention, but chemotherapy.

ODC regulations are delicately by TPA, cGMP and FBS (growth factors), etc. ODC, antizyme (AZ) and antizyme inhibitor (AZI) are fickle proteins and all regulate polyamines. Authors did not discuss anymore.

Ornithine decarboxylase (ODC) is a rate-limiting enzyme in polyamine biosynthesis. ODC protein degraded by ubiquitin-independent (AZ-dependent) pathway, but it also theatrical increase following polyamine decline! ODC half-life is only 30 mins and it is delicately regulated by AZ and AZI (also short half-life). DMFO is suicide inhibitor of ODC, therefore ODC is not worthy therapeutic target. Authors emphasize on polyamine is well. But there is no any discussion between on ODC/AZ/AZI regulation. It’s weakness on the “Title: Regulation of Polyamine Metabolism by Curcumin for Cancer Prevention and Therapy”.

ODC is tumor promotion factor (cellular growth), but author need to define it on the roles of tumor initiation (mutation by carcinogens or mutagens) and programming. If ODC and polyamine were not, the therapeutic prediction is weak.

Different condition (time, location and concentration) of curcumin have diverse effects, curcumin own the anti-oxidation but it also increase ROS and apoptosis. There are two important things need to assess. The first, whether ODC or polyamine have normal function, such as benefit to protect aging? ODC and polyamines are not all bad effect. They could protect drugs- or ROS-induced cell insults on normal cells such as brain and neurons. Authors need to emphasize this! The second, readers need to realize the real role of curcumin-interfered with ODC. Is it directly or indirectly effect on carcinogenesis?

The readers need authors draw more beautiful and color Figures.

Increase the Figures of ODC/AZ/AZI regulation, discussion and references.

Authors need more discussion at the feasibly molecular mechanism on the interference of ODC therapeutic evidences by curcumin and both of the benefit and disadvantage application.

Finally, “Polyamine Metabolism in Disease and Polyamine-Targeted Therapies” this topic need to discuss very carefully. ODC and polyamine have dramatic regulation and they are not bad in everything. Be care!

Reviewer 3 Report

Regulation of Polyamine Metabolism by Curcumin 3 for Cancer Prevention and Therapy 4 Tracy Murray-Stewart and Robert A. Casero, Jr 

The review presented by  Tracy Murray-Stewart and Robert A. Casero, Jr concerns the modulation of polyamine metabolism by curcumin and its potential application in chemopreventive and chemotherapeutic  approach. The review reports many studies on the subject and in my opinion is comprehensive and well written. However I have only a small observation and a suggestion, briefly exposed in the following point:

 As mentioned by the authors, despite the broad effects of curcumin on the biological functions of cells, its potential use as a therapeutic agent is severely affected by its low water solubility, poor in vivo bioavailability, and rapid metabolism. In recent years, its structure has been widely modified, focusing mainly on changes in the β-diketone structure and aryl substitution pattern of the molecule. Among the obtained analogues, the 3,5-dibenzylidenepiperidin-4-one framework was found particularly interesting because of its higher antioxidant and antiproliferative actions with respect to curcumin.. This analogue may also offer the chance of an improved pharmacokinetic profile. According to these information I would suggest to authors to insert the following references:Youssef, K. M.; El-Sherbeny, M. A.; El-Shafie, F. S.; Farag, H. A.; Al-Deeb, O. A.; Awadalla, S. A. Synthesis of curcumin analogues as potential antioxidant, cancer chemopreventive agents. Arch. Pharm. (Weinheim, Ger.) 2004, 337, 42–54.

Pati, H. N.; Das, U.; Quail, J. W.; Kawase, M.; Sakagami, H.; Dimmock, J. R. Cytotoxic 3,5-bis(benzylidene)piperidin-4-ones and N-acyl analogs displaying selective toxicity for malignant cells. Eur. J. Med. Chem. 2008, 43, 1–7.

Padhye, S.; Chavan, D.; Pandey, S.; Deshpande, J.; Swamy, K. V.; Sarkar, F. H. Perspectives on chemopreventive and therapeutic potential of curcumin analogs in medicinal chemistry. Mini-Rev. Med. Chem. 2010, 10, 372–387. I would like to point out that these three citations are not present in: Adiwidjaja, J.et al. 2017

Author Response

Thank you for your review and suggestion to include additional references to curcumin analogues. Text and citations were added to Section 5, lines 386-391.